# Accepting a "New" Standard Variety: Comparing Explicit Attitudes in Luxembourg and Belgium

**Judit Vari * and Marco Tamburelli** 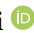

Department of Linguistics, English Language and Bilingualism, Bangor University, Bangor LL57 2DG, UK; m.tamburelli@bangor.ac.uk
* Correspondence: J.Vari@bangor.ac.uk

**Abstract:** Language maintenance efforts aim to bolster attitudes towards endangered languages by providing them with a standard variety as a means to raise their status and prestige. However, the introduced variety can vary in its degrees of standardisation. This paper investigates whether varying degrees of standardisation surface in explicit attitudes towards standard varieties in endangered vernacular speech communities. Following sociolinguistic models of standardisation, we suggest that explicit attitudes towards the standard variety indicate its acceptance in vernacular speech communities, reflecting its overall degree of standardisation. We use the standardised Attitudes towards Language (AtoL) questionnaire to investigate explicit attitudes towards the respective standard varieties in two related vernacular speech communities—the Belgische Eifel in Belgium and the Éislek in Luxembourg. The vernacular of these speech communities, Moselle Franconian, is considered generally vulnerable (UNESCO), and the two speech communities have opted to introduce different standard varieties: Standard Luxembourgish in Luxembourg shows lower degrees of standardisation and is only partially implemented. In contrast, Standard German in the Belgian speech community is highly standardised and completely implemented. Results show that degrees of standardisation surface in speakers' explicit attitudes. Our findings have important implications for the role of standardisation in language maintenance efforts.

**Keywords:** standardisation; language attitudes; Moselle Franconian



## 1. Introduction

Language maintenance efforts aim to bolster the vitality of endangered languages through a number of interventions, often including the introduction of a standard variety into the endangered speech community (e.g., Grenoble and Whaley 2005; Lane et al. 2018). It is generally agreed within language maintenance research that the prestige and functions associated with a standard variety benefit its endangered vernaculars by improving attitudes which, in turn, bolsters usage and consequently vitality (Fishman 1991, 2001; Lewis and Simons 2010). More specifically, researchers show that the introduction of a standard variety leads to use of the endangered language in more language domains overall and especially more prestigious domains such as education (Loureiro-Rodriguez et al. 2013; O'Rourke 2010). The additional functions and prestige of the standard variety are seen as a positive influence on the perception of the endangered vernaculars which are subsequently viewed as being part of a language in its own right (Fishman 1991).

The underlying assumption behind this claim is that the newly introduced standard variety will carry prestige. This assumption is corroborated by an abundance of studies showing that speakers hold more positive attitudes towards a standard variety compared to its vernacular (Giles and Marlow 2011; Milroy 1991; Preston 1989; Rosseel et al. 2018).

These studies find favourable evaluations of a standard variety by investigating two different types of attitudes, i.e., implicit and explicit. This distinction is mainly based on the criteria of awareness, but also on criteria from social cognition, including the concept of

automaticity (De Houwer and Moors 2007; Rosseel and Grondelaers 2019). In this paper, we follow the latter approach, which defines explicit and implicit attitudes based on systematic vs. automatic underlying processes of social cognition depending on whether they require higher or lower degrees of cognitive resources and time (Fazio and Towles-Schwen 1999; Wilson et al. 2000).

Besides the general agreement on the prestige of a standard variety, research identifies varying degrees of standardisation (Coupland and Kristiansen 2011; Ferguson 1968; Haugen 1966). The most well-known sociolinguistic framework for the different stages of standardisation is probably Haugen's model (Haugen 1966), which identifies four stages: norm selection; codification; elaboration of functions; and acceptance by the speech community, with researchers identifying this last stage as crucial in the standardisation process (Ammon 1989; Coupland and Kristiansen 2011; Haugen 1966, 1997). More specifically, Haugen (1966) argues that the last stage is critical, since it ultimately leads to the implementation of the standard variety and its functions in the community. In addition, studies identify positive attitudes towards the standard variety, i.e., its prestige, to indicate its acceptance in the speech community (Devonish 2003; Mattheier 2003; De Groof 2002; Feitsma 2002).

Prestige is occasionally thought to comprise two different attitude dimensions: on the one hand, covert prestige, which touches on aspects of dynamism and solidarity and, on the other hand, traditional overt prestige tied to status and domination (Cargile et al. 1994; Grondelaers and van Gent 2019; Grondelaers et al. 2016). Further, some studies identify the emergence of new types of standard varieties based on the prestige of "media cool" (Grondelaers et al. 2016, p. 134). On the contrary, the current study does not further distinguish between different types of prestige, since our speech communities motivate a more generalized approach towards prestige (see Sections 1.1 and 1.2). More specifically, prestige is not necessarily based on dynamism aspects due to the rural context of our speech communities and the absence of the vernacular in the media. In addition, no studies so far have explored social identity in our speech communities, thus making it difficult to investigate the solidarity dimension of attitudes. Overall, our definition of attitudes, thus prestige, focuses rather on the underlying cognitive processes than different types of content.

Overall, varying degrees of standardisation and the associated variation in prestige imply that there might be limitations to the positive effect of a standard variety. If a standard has not reached the last stage of acceptance in the speech community, it might not hold the prestige that language maintenance researchers argue will complement its endangered vernaculars. Thus, an investigation of how well a standard variety is accepted in the community is the first step before any potentially positive effect on its vernaculars can be explored. Indeed, acceptance of a newly introduced standard has been the subject of numerous studies in language maintenance research (Devonish 2003; O'Rourke 2018; Urla et al. 2018). However, very few studies investigate this dimension in relation to attitudes (Urla et al. 2018; O'Rourke 2010), or compare closely related varieties that differ in degree of standardisation. Such comparative attitudinal studies would provide insights into the trajectory of standardisation processes and, therefore, into acceptance of the standard. Overall, such insights are necessary to fully understand the workings of standardisation and its potential contribution to language maintenance efforts.

In the following, we contribute to filling this research gap by presenting a comparative study of two speech communities with related endangered vernaculars and standard varieties. We selected Canton *Clervaux* (Luxembourg) and the *Belgische Eifel* (Belgium) for three reasons. Firstly, the Moselle Franconian vernaculars of these speech communities are linguistically very closely related (Bruch 1953; Mattheier and Wiesinger 1994; Wiesinger 1982a, 1982b) and they are considered to be vulnerable (UNESCO 2017).

In addition, in both speech communities, the vernaculars are in contact with additional standardised varieties (French in Belgium, French and German in Luxembourg) besides their respective standards.

Importantly, however, the two communities have opted for different ways of introducing a standard variety: in Luxembourg, the Moselle Franconian speakers have an "own" endogenous standard (i.e., Standard Luxembourgish), whereas in Belgium, the Moselle Franconian vernaculars are associated with an exogenous standard, namely standard German. These two types of standardisation led to varying degrees of linguistic distance (i.e., Abstand in the sense of Kloss (1978)) between the endangered vernaculars and their standard, while also leading to varying degrees of standardisation (i.e., Ausbau in the sense of Kloss (1978)). This paper focuses on this latter point, namely the varying degrees of standardisation, with the aim to investigate how different degrees of standardisation resulting in different degrees of acceptance may surface in different attitudes across two speech communities.

The following section discusses language attitudes and standardisation processes in the two speech communities in order to establish the different degrees of standardisation of their respective standard varieties, i.e., Standard German and Standard Luxembourgish. Particular attention is given to the final stage of standardisation, namely acceptance (Haugen 1966), with the intention of determining whether previous attitudinal studies found any differences across the two speech communities.

### 1.1. Belgische Eifel/"Deutschsprachige Gemeinschaft" in Belgium

The first Moselle Franconian speech community under investigation is situated in the southern part of a political unit called the *Deutschsprachige Gemeinschaft* ('German speaking community')[1] in Belgium.

The territory of the modern day *Deutschsprachige Gemeinschaft* was ceded to Belgium by the German Empire in 1919. The highly autonomous *Deutschsprachige Gemeinschaft*, where Standard German is the only official language, has legislative and executive powers similar to the French and Dutch speaking communities and to the bilingual area around Brussels-Capital. Overall, German is one of the three official languages of Belgium (alongside French and Dutch) and the approximately 70,000 German speakers constitute the smallest speech community in Belgium, totalling only around 0.6% of the Belgian population (Möller 2017). Additionally, the *Deutschsprachige Gemeinschaft* is part of the French speaking Walloon region of Belgium, on which it depends both economically and politically (Combuchen 2009; Möller 2017).

The *Deutschsprachige Gemeinschaft* community lacks an "own" endogenous standard, since it associates its Germanic vernaculars (including Moselle Franconian) with an exogenous standard variety, namely Standard German (Möller 2017). The standardisation processes of German are at a very advanced stage and their beginnings can be dated back to at least the 16th century (Mattheier 2003). Some have argued that the degree of codification of Standard German is higher compared to some other highly standardised varieties, e.g., English, since it even includes codification of a spoken standard (Durrell 1999; Ferguson 1968). Its functions are highly elaborated for usage in different contexts in its "own" speech community in addition to functions and prestige in an international context (Ammon 2015; Mattheier 2003). Consequently, attitudes towards Standard German are shown to be overwhelmingly positive when compared to other standardised languages present in Germany, such as English, Turkish and French (Rothe 2012; Schoel et al. 2012a). Similarly, studies show more positive attitudes towards Standard German in contrast to its vernaculars in Germany and outside, e.g., in autochthon minority communities (Adler 2019; Deminger 2000; Schoel and Stahlberg 2012). Finally, the prestige of Standard German is intertwined with high levels of prescriptivism and linguistic discrimination against vernaculars and regional variation (Davies 2006; Maitz and Elspaß 2012; Schmidlin 2011).

In the *Deutschsprachige Gemeinschaft*, Standard German is well-implemented in all domains. Sociolinguistic analyses show that it covers functions such as school, media, and administrative use (Ammon 1995; Combuchen 2009; Nelde and Darquennes 2002). More specifically, Standard German has a major role in the education system, both as the medium



of instruction and as a school subject (Combuchen 2009) which, research suggests, influences attitudes significantly (Davies 2018; Horner and Weber 2015; Woolard and Gal 2001).

While empirical studies on speakers' perception of the varieties present in the *Deutschsprachige Gemeinschaft* are extremely scant (Gramß 2008; Weber 2009), there is some evidence that Standard German is accepted at a contextual level, with participants reporting it as obligatory in language domains such as work, government and education. Similarly, participants recognise model speakers such as local politicians and news presenters, who they rate as speaking more "standard-like" (Weber 2009).

There is only one large-scale quantitative study comparing explicit attitudes towards Standard German and Germanic vernaculars in the *Deutschsprachige Gemeinschaft*, i.e., (Weber 2009). Participants were asked to indicate in a questionnaire which variety they preferred or whether they equally liked both. Overall, results show predominantly egalitarian explicit attitudes across the *Deutschsprachige Gemeinschaft*. Accordingly, most participants from the *Belgische Eifel* region of the *Deutschsprachige Gemeinschaft* report equally liking Standard German and their Moselle Franconian vernacular, but a sizeable minority of respondents from this region prefer Moselle Franconian over Standard German. Typically, participants showed overwhelmingly positive attitudes towards vernaculars on the solidarity dimension, i.e., integrative attitudes. This attitude dimension is believed to index social identity and a feeling of belonging (Cargile et al. 1994; Lambert et al. 1965; Ryan Bouchard et al. 1982). Additionally, the study found somewhat positive attitudes towards Standard German on the status/instrumental attitude dimension, which is indicative of social, political and economic status (Cargile et al. 1994; Lambert et al. 1965; Ryan Bouchard et al. 1982).

Besides the significant lack of quantitative studies on explicit attitudes, we are only aware of one study investigating implicit attitudes in the *Deutschsprachige Gemeinschaft* (Vari and Tamburelli 2020). Their method applied an *Implicit Association Test* based on Greenwald et al. (1998). This study did not find the same egalitarianism in implicit attitudes that the explicit attitude study demonstrated (Weber 2009), reporting instead that implicit attitudes towards Standard German were more positive compared to its Moselle Franconian vernaculars in the *Belgische Eifel* region. This is in line with sociolinguistic and social psychological research, where speakers are shown to be less likely to correct implicit attitudes according to social expectations and official ideologies as they do for explicit attitudes (Dovidio et al. 2009; Fazio and Towles-Schwen 1999; Kristiansen 2015; Wilson et al. 2000), and with sociolinguistic studies arguing that a highly standardised variety tends to carry heightened prestige (Coupland and Kristiansen 2011; Haugen 1966). However, some difficulties arise when contextualising these studies on attitudes in the *Deutschsprachige Gemeinschaft*. In sociolinguistic research, explicit and implicit attitudes towards standard and vernacular varieties vary depending, on the one hand, on definitions of explicitness vs. implicitness and additionally, on the type of vernacular including, for example, low register urban varieties as well as rural traditional dialects (e.g., Deminger 2000; Kristiansen 2015; Rosseel et al. 2019).

Neither of the two attitudinal studies reported above measured attitudes (explicit or implicit) towards Standard German and its vernaculars in relation to French, an additional standardised contact variety present in the region (Vari and Tamburelli 2020; Weber 2009). Thus, it remains unclear to what degree French might also carry prestige and perhaps even be a functional standard variety[2] for the vernaculars of the *Deutschsprachige Gemeinschaft*, with some studies reporting that it competes with Standard German over H(igh) domains in some parts of the region (Gramß 2008; Nelde and Darquennes 2002). French is found to be present next to Standard German in the work sphere and employment is also sought in neighbouring countries such as Germany, France and Luxembourg as well as in the francophone regions of Belgium (Gramß 2008; Möller 2017; Nelde and Darquennes 2002). Overall, this multilingual contact situation has led to some degree of language endangerment, with Moselle Franconian vernaculars being identified as vulnerable varieties (UNESCO 2017).[3]

The southern region of the *Deutschsprachige Gemeinschaft*, i.e., the *Belgische Eifel* (see Figure 1 below) has the most widespread usage and the highest levels of Moselle Franconian competence within the *Deutschsprachige Gemeinschaft* (Darquennes 2019; Nelde and Darquennes 2002; Weber 2009), making it particularly relevant to our study, as both competence and usage are well known to influence attitudes (Garrett 2010; Lambert et al. 1968).

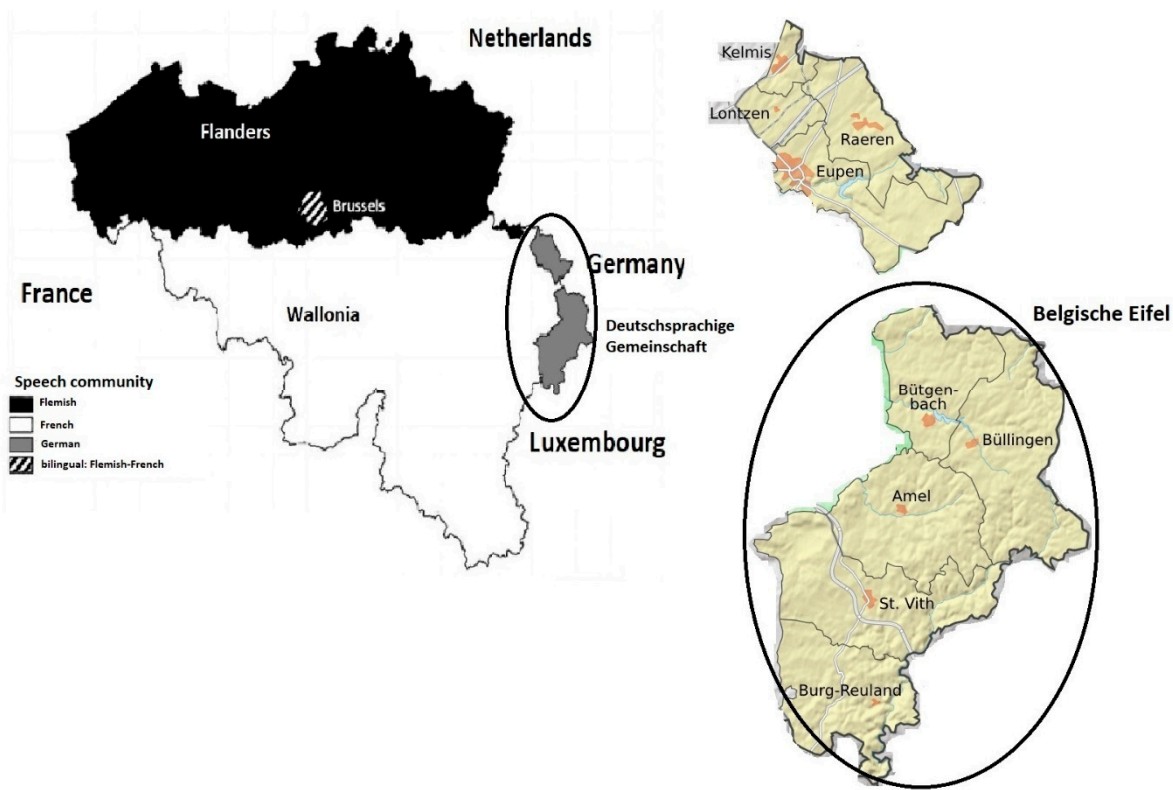

**Figure 1.** The location of the Belgische Eifel, based on (Verhiest 2015, p. 55).

The *Belgische Eifel* constitutes five districts of the *Deutschsprachige Gemeinschaft*, namely, Amel, Büllingen, Burg-Reuland, Bütgenbach and St. Vith, and is a predominantly rural area with its 631 km$^2$ and a population of 30,219. Dialectological studies report that the majority of Moselle Franconian vernaculars in this area are closely related to the Moselle Franconian spoken in the Éislek region of Luxembourg (Bruch 1953; Mattheier and Wiesinger 1994; Wiesinger 1982a, 1982b).

### 1.2. Clervaux/the Éislek region of Luxembourg

The second speech community under investigation is situated in Luxembourg, where Moselle Franconian varieties have undergone some standardisation (Gilles 2015; Newton 2000; Stell 2006) since the establishment of Luxembourg as an independent nation state in 1839. This standardisation involved the creation of a "new", endogenous standard variety, namely Luxembourgish/*Lëtzebuergesch* (Stell 2006). Historically considered a German "dialect", Luxembourgish was originally only a spoken variety, used mainly in the home domain (Gilles 2019; Newton 1996; Stell 2006). During the 19th century, a written tradition of Moselle Franconian developed in Luxembourg, even if clearly considered to be "only" folk literature in the vernacular (Gilles 2019). Finally, standardisation processes resulted in codification, including, for example, the development of the *Lëtzebuerger Online Dictionnaire*, and an increase in model texts since the 1980s (Gilles 2019; Stell 2006). Luxembourgish was recognised as a national language in 1984 (alongside French and German as official languages[4]) and its functions are now significantly more elaborated compared to its originally exclusive use in the home domain. Today, Luxembourgish occupies main functions in the

political sphere and in the (digital) media. Overall, Luxembourgish is now considered an *Ausbau* language—in the sense of Kloss (1978)—with around 266,000 native speakers and a significant number of L2 learners (Fehlen 2016; Weber-Messerich 2011).

However, researchers have argued that the standardisation processes are not complete since Standard Luxembourgish has not reached the last stage of full implementation (see for example Gilles 2015). Typically, Standard Luxembourgish only plays a minor role in the education system, resulting in limited implementation of existing codification, for example, spelling norms (Gilles 2015; Horner and Weber 2010). Despite its occasional, unofficial use in the classroom (Redinger 2010), Standard Luxembourgish is not the official medium of instruction and its teaching as an L1 is limited (Horner and Weber 2015). Additionally, a lack of prescriptivism could also be indicative of the limited implementation of Standard Luxembourgish in the speech community. Typically, teachers are advised by the ministry of education to demonstrate high levels of tolerance regarding spelling norms (Horner and Weber 2010). This officially endorsed linguistic tolerance suggests that Standard Luxembourgish has lower levels of prescriptivism compared to other standard varieties, such as Standard German (Davies 2006; Horner and Weber 2015).

A variety's degree of standardisation in a speech community also shows in the standard variety's level of acceptance by speakers (Haugen 1966). Attitudinal studies yield conflicting results regarding how well Standard Luxembourgish is accepted as a prestigious standard (Bellamy and Horner 2018; Entringer et al. 2018; Fehlen 2009; Gilles et al. 2010; Neises 2013). On the one hand, a qualitative study found that speakers doubt whether Luxembourgish can be considered a fully fledged language, especially in comparison with other highly standardised varieties such as German and French (Bellamy and Horner 2018). On the other hand, perceptual studies demonstrate an awareness among speakers of the contexts in which Standard Luxembourgish is used in model texts, e.g., invitations to official events, or by model speakers, e.g., news presenters (Entringer et al. 2018; Fehlen 2009; Neises 2013).

Generally, norm awareness can also be present at a geographical level, when speakers localise a region of the standard variety, for example, the "Copenhageness of Danish" (Kristiansen and Jaworski 1997). Numerous perceptual dialectological studies show that this localisation of a standard variety also surfaces in attitudes towards regional variation (Eichinger and Stickel 2012; Preston 1989, 1999). However, to the best of our knowledge, only three quantitative studies have investigated attitudinal differences between Moselle Franconian varieties in Luxembourg (Entringer et al. 2018; Neises 2013; Vari and Tamburelli 2020). These have shown that speakers typically identify varieties of the Alzette Valley and Luxembourg City as the most "standard-like" in contrast to the varieties from the northern Éislek region, especially the Canton Clervaux, which are perceived to be the most "non-standard-like". Similarly, speakers hold more positive explicit attitudes towards the "standard-like" varieties than towards varieties spoken in the Éislek region, or specifically, Clervaux (Entringer et al. 2018; Neises 2013). This difference also shows in explicit attitudes towards speakers of these varieties (Neises 2013), especially in relation to traits such as intelligence, social status and correctness, which are indicative of a standard speaker (Milroy 1991). However, participants in one of these studies were likely to have come predominantly from the "standard region", i.e., the Alzette Valley, themselves (Neises 2013), and thus likely to evaluate their own variety positively. The second study did not include information regarding participants' region of provenance in the results (Entringer et al. 2018).[5]

We are only aware of one study that investigated attitudes exclusively in the Éislek region, and specifically Canton Clervaux (Vari and Tamburelli 2020), whose vernacular speakers are identified as the most "non-standard-like" (Entringer et al. 2018; Neises 2013). This quantitative study explored vernacular speakers' implicit attitudes, which have been demonstrated to be less influenced by social desirability and official ideology (Dovidio et al. 2009; Fazio and Towles-Schwen 1999; Kristiansen 2015; Wilson et al. 2000). Results showed more positive implicit attitudes towards the Moselle Franconian vernacular of this region compared to Standard Luxembourgish.

The geographical localisation of standard and vernacular regions, which was reported in perceptual studies, is in line with dialectological and phonological studies (Bruch 1953; Gilles 1999). The varieties in the Éislek region are reported to constitute a separate dialect area, which retains most regional features and differs markedly from the varieties in the Alzette valley (Entringer et al. 2018; Gilles 1998; Gilles and Trouvain 2013). The Moselle Franconian of the Éislek region—especially the vernacular of the most northerly part, namely Canton Clervaux—is closely related to Moselle Franconian in the *Belgische Eifel* (Bruch 1953; Mattheier and Wiesinger 1994; Wiesinger 1982b).

Canton Clervaux, with a size of 342 km$^2$ and a population of 18,436 (STATEC 2019a, 2019b), is situated in Luxembourg's northern, rural border region, neighbouring Belgium and Germany (see Figure 2 below). It has five districts: Parc Hosingen, Wincrange, Troisvierges, Weiswampach and the city of Clervaux itself. However, dialectological studies exclude Parc Hosingen from a more or less homogenous northern dialect area (Bruch 1953; Gilles 1999), a stance also taken by the only attitudinal study of this region (Vari and Tamburelli 2020). Information about the usage of Moselle Franconian vernacular and the competence of its speakers in Canton Clervaux is scarce. Studies establishing speaker numbers of Luxembourgish often lack the distinction between vernacular and Standard Luxembourgish/Moselle Franconian, for example, Fehlen (2016). However, in a large-scale study (Fehlen 2009), 50% of the participants from Canton Clervaux considered themselves to be vernacular Moselle Franconian speakers.

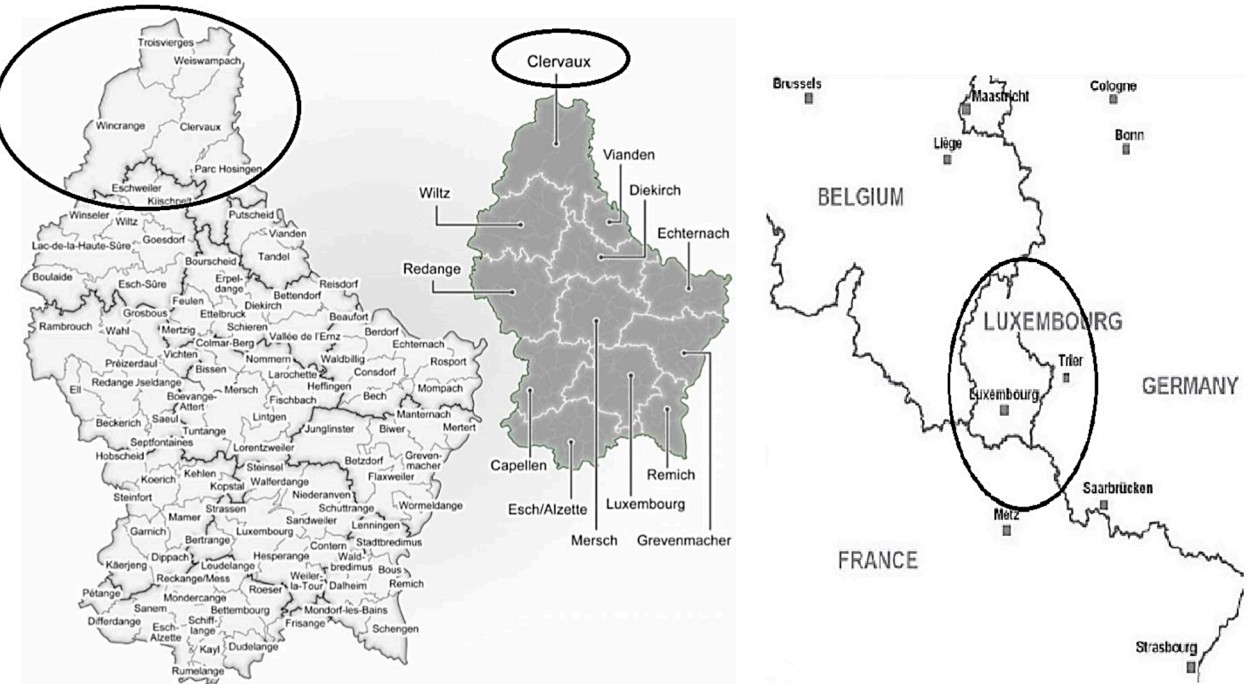

**Figure 2.** Canton Clervaux, situated in Luxembourg (Neises 2013; STATEC 2019a).

The sociolinguistic situation in Luxembourg, including Canton Clervaux, is characterised by high degrees of multilingualism. Historically, Luxembourgish has been in contact with Standard German and French, which have occupied H(igh) domains for a longer time and more extensively than the newly standardised Luxembourgish (Gilles 2019; Horner and Weber 2008; Newton 1996). Standard German in particular acted as a structural standard variety for the Moselle Franconian speech community during the 19th and 20th century, before Standard Luxembourgish was introduced (Gilles 2019; Stell 2006; Ziegler 2012). In addition, French historically occupied functions of a standard variety and is identified as potentially contributing to the endangerment of the Moselle Franconian vernaculars (Fehlen 2016; UNESCO 2017).

Research shows conflicting findings regarding how well Standard German and French are implemented and accepted in modern-day Luxembourg. Both contact varieties still occupy H(igh) domains such as the media and the workplace. More specifically, around half of the Luxembourgish nationals report regularly using French and German at the workplace (Fehlen 2016). However, their acceptance as prestigious standard varieties among Luxembourgish speakers is mixed, though studies on attitudes towards Luxembourgish in relation to other standardised contact varieties are scant. Only two quantitative studies explored explicit attitudes towards Luxembourgish in relation to French and German (Fehlen 2009; Gilles et al. 2010), while one study investigated only explicit attitudes towards Luxembourgish in relation to French (Lehnert 2018). In two of these studies (Fehlen 2009; Lehnert 2018), speakers slightly preferred French over Luxembourgish, whereas the study by Gilles et al. (2010) found Luxembourgish to be ranked first, followed by French in second place, and German in third place. Fehlen (2009) also found German to be the least favourable language.

These conflicting findings regarding the most preferred language variety, i.e., French vs. Luxembourgish, might be a result of methodological differences, such as different semantic differential scales (i.e., modern, useful, pleasant) of the questionnaires. Only Lehnert (2018) used the Attitudes towards Language (AtoL) questionnaire (Schoel et al. 2012b), one among numerous standardised questionnaires in language attitudes research (Giles and Rakić 2014; Mulac and Lundell 1982; Zahn and Hopper 1985). By using the AtoL questionnaire, Lehnert (2018) aimed to measure language attitudes exclusively, as opposed to speaker evaluation. Attitudinal studies found evidence that these two concepts might differ, even if they are often mixed together (Cargile et al. 1994; Gilles et al. 2010; Lehnert 2018; Neises 2013). In addition, Lehnert (2018) complemented her explicit attitude measure with an implicit attitude measurement. This implicit attitude measurement has been shown to be less influenced by social desirability and official ideology (Dovidio et al. 2009; Fazio and Towles-Schwen 1999; Kristiansen 2015; Wilson et al. 2000), and its application revealed a preference for Luxembourgish over French, unlike in the explicit attitude measure.

To the best of our knowledge, no quantitative studies have yet investigated explicit attitudes towards Standard Luxembourgish and Moselle Franconian vernaculars in relation to additional standardised contact varieties, i.e., French and German. The studies reported above (Fehlen 2009; Gilles et al. 2010) considered Luxembourgish as a presumed homogenous entity, failing to distinguish between the standard variety, i.e., Standard Luxembourgish, and its Moselle Franconian vernaculars, all of which are part of what is considered "the Luxembourgish language". Regional variation of Luxembourgish was also not considered in these studies. Participants were all reported to be Luxembourgish nationals, occasionally contrasted with non-nationals (Fehlen 2009; Gilles et al. 2010), but no distinction was made regarding their residence or origin within Luxembourg. Therefore, we suggest that these attitudinal studies cannot be assumed to be necessarily indicative of explicit attitudes towards the Moselle Franconian vernacular spoken in Canton Clervaux, in the northern vernacular region of the Éislek.

The next section outlines the research questions and hypotheses that underlie the present study. For the sake of brevity, the Moselle Franconian speech communities under investigation, i.e., Canton Clervaux in the Éislek region and the *Belgische Eifel* in the *Deutschsprachige Gemeinschaft*, will henceforth be referred to as 'Luxembourg' and 'Belgium', respectively.

### 1.3. Research Questions and Hypotheses

This study aims to investigate the final stage of a variety's standardisation: acceptance in speech communities, especially in endangered speech communities. Previous research suggests that speakers' attitudes reflect how well a standard variety is implemented and accepted in any speech community, endangered or otherwise (Devonish 2003; O'Rourke 2018; Urla et al. 2018; De Groof 2002; Feitsma 2002).

The Moselle Franconian communities of Belgium and Luxembourg lend themselves to an investigation of the relationship between standardisation and speakers' attitudes, since their respective standard varieties, i.e., Standard Luxembourgish and Standard German, vary in their degree of standardisation. As emerged from the literature review, Standard German is a highly standardised variety with elaborated functions in Belgium. In contrast, the standardisation processes of Luxembourgish are incomplete, with Standard Luxembourgish lacking certain functions, for example, in the educational domain.

This paper focuses on the explicit level to investigate how degrees of standardisation emerge in attitudes in both Moselle Franconian speech communities. Social psychological research shows that explicit and implicit attitudes can influence each other and only an investigation of both attitude types allows us to fully understand the evaluation of objects, people and events (Gawronski et al. 2009; Whitfield and Jordan 2009; Wilson et al. 2000). Accordingly, we suggest that both types of attitudes reflect how well a standard variety is accepted in an endangered speech community. A study of explicit attitudes is not only needed to complement insights from implicit attitudes (Vari and Tamburelli 2020), but it is specifically important in the special context of our Moselle Franconian speech communities. The ongoing standardisation processes in Luxembourg motivate an outlook on attitudes rather than only a snapshot of current attitudes. Numerous studies find that explicit attitudes are the "window into the future", arguing that attitude change manifests first in explicit attitudes (McKenzie and Carrie 2018; Dovidio et al. 2009; Wilson et al. 2000). Standardisation in Luxembourg is arguably nowadays more hegemonic in its nature based on top-down language policies rather than grass-root movements, which researchers occasionally identify in modern day minority language communities (Costa et al. 2018). We suggest that especially top-down standardisation, such as in Luxembourg, is more likely to manifest first in conscious propositional learning processes, which social psychological studies find influence mainly explicit attitudes (Gawronski et al. 2009). Consequently, we aim to investigate explicit attitudes towards Standard German and Standard Luxembourgish in relation to their Moselle Franconian vernaculars.

Overall, we aim to explore the following research question:

(a)    Are explicit attitudes towards Standard German in Belgium more positive than towards Standard Luxembourgish in Luxembourg, as suggested by their different degrees of standardisation?

We suggest that explicit attitudes towards the respective standard variety, i.e., Standard German or Standard Luxembourgish, will indicate its acceptance in Luxembourg or Belgium, reflecting its overall degree of standardisation.

We hypothesise that these different degrees of acceptance will surface in explicit attitudes in (1) within- and (2) between-speech community comparisons and therefore,

**Hypothesis 1 (H1).** *Luxembourgish speakers will hold more **negative** explicit attitudes towards their respective standard variety, i.e., Standard Luxembourgish, compared to Moselle Franconian vernaculars. Conversely, Belgian speakers will hold more **positive** explicit attitudes towards their respective standard variety, i.e., Standard German, compared to Moselle Franconian vernaculars.*

**Hypothesis 2 (H2).** *Luxembourgish speakers will hold more **negative** explicit attitudes towards their standard variety, i.e., Standard Luxembourgish, compared to Belgian speakers' explicit attitudes towards their own standard variety, i.e., Standard German.*

Social psychological research implies that our hypotheses regarding explicit attitudes in Belgium and Luxembourg need to be independent from the findings of the previous comparative study on implicit attitudes. More specifically, explicit attitudes might potentially be subject to more influence from social desirability in the form of official ideologies compared to implicit attitudes (Dovidio et al. 2009; Fazio and Towles-Schwen 1999; Kristiansen 2015; Wilson et al. 2000). However, there is no information on whether and how social desirability might influence explicit attitudes towards the respective standard

varieties when compared to implicit attitudes. Specifically, research shows that the way social desirability influences explicit attitudes is dependent on the socio-political and cultural context of the participants (Dovidio et al. 2001), but attitudinal studies in Luxembourg and Belgium are scarce and show mixed results (Fehlen 2009; Gilles et al. 2010; Gramß 2008; Lehnert 2018; Weber 2009).

In addition, based on findings from language maintenance research, we expect that whether speakers accept the respective standard variety also depends on other standardised contact varieties present in the speech community (Fishman 1991, 2001). Very positive attitudes towards other standardised contact varieties have been shown to negatively influence the acceptance of a "new" standard (Loureiro-Rodriguez et al. 2013; O'Rourke 2018). However, no study has thus far investigated attitudes towards Moselle Franconian vernaculars or their respective standard variety in relation to other standardised contact varieties, namely German and French. Therefore, this study also explores explicit attitudes toward other standardised contact varieties in Luxembourg and Belgium by addressing the following research question:

(b)　What are the explicit attitudes towards additional standardised contact varieties, i.e., French in Belgium, and French and German in Luxembourg?

Particularly in Luxembourg, German could impede the acceptance of Standard Luxembourgish, due to its former role as a structural standard variety for the Moselle Franconian vernaculars of Luxembourg (Gilles 2019). However, the few attitudinal studies conducted in Luxembourg found Standard German to have low prestige (Fehlen 2009; Gilles et al. 2010), despite its still widespread usage in the media and in the education system (Fehlen 2016; Wagner 2015). This mismatch between high levels of usage and low levels of prestige prevents us from presenting a hypothesis, as does the complete lack of quantitative attitudinal research in Belgium comparing French and Standard German. Therefore, the investigation of attitudes towards additional standardised contact varieties remains exploratory in nature.

## 2. Materials and Methods

### 2.1. Participants

Participants were recruited via advertisement in the local media and cooperations with local societies in the speech communities such as local choirs and women's clubs. Overall, 167 participants took part in the present study, but only 131 were included in the final analysis. We excluded participants below 20 and above 60 years of age in order to ensure more homogenous samples regarding age and speech community. This resulted in a more balanced design in contrast to the originally larger Belgian sample (90 participants) compared to the Luxembourgish sample (77 participants). A balanced design facilitates statistical analysis (Field 2009).

Finally, the Luxembourgish sample included 62 participants (38 females, 24 males, mean age = 35.7 years, s.d. = 12.1). Overall, participants assessed themselves as highly competent in all varieties under investigation on a 5-point Likert scale (from 0/not at all, to 4/perfect: mean = 3.23, s.d. = 0.47). The ratings of their language competences differed significantly (Friedman's ANOVA: $\chi^2(3) = 83.3$, $p < 0.01$) such that they reported their French competence to be the lowest (mean = 2.68, s.d. = 0.68) and their vernacular competence to be the highest (mean = 3.71, s.d. = 0.56).

In Belgium, 69 participants (43 females, 26 males, mean age = 40.3 years, s.d. = 10.4) took part in the study. Their overall self-assessed language competence was also high (mean = 3.06, s.d. = 0.48) and their language competence in the three varieties differed significantly (Friedman's ANOVA: $\chi^2(2) = 73.0$, $p < 0.01$). They also rated their French competence to be the lowest (mean = 2.53, s.d. = 0.68). However, unlike their Luxembourgish counterparts, they reported the highest competence in their standard variety, Standard German (mean = 3.41, s.d. = 0.52). Nevertheless, the vernacular competence of the Belgian participants was still fairly high (mean = 3.23, s.d. = 0.71) and comparable to that of their Luxembourgish counterparts.

Unfortunately, participants' socio-economic status was not recorded, due to a technical error. However, we suggest that French competence could partially indicate participants' socio-economic status. Research on Luxembourgish speakers found that their competence in French correlated moderately with educational attainment (Fehlen 2009, 2016), due to the importance of French in secondary and higher education in Luxembourg. In Belgium, there is no such relationship between educational attainment and competence in French, due to the speech community's different education system and socio-political background.

All participants were Luxembourgish and Belgian nationals, respectively, and reported to have spent the majority of their childhood living in their respective speech community, as described in Sections 1.1 and 1.2.

*2.2. The Attitudes towards Language (AtoL) Questionnaire*

To investigate our hypotheses and to measure explicit attitudes, we used a multi-scale online questionnaire with semantic differential scales featuring bipolar adjectives (Osgood 1952). Two reasons motivated our decision against applying the speaker evaluation paradigm, i.e., Matched or Verbal Guise Experiments (Lambert et al. 1960; Ryan Bouchard and Carranza 1977). First, there is significant controversy regarding whether such experiments constitute a measure of explicit attitudes, due to the fact that they involve partial deception, and depending on how one approaches the distinction between explicit and implicit (Adams 2019; Kristiansen 2015; Rosseel and Grondelaers 2019). On the other hand, it is generally agreed that survey studies specifically measure explicit attitudes because they present participants directly with overt questions regarding their preferred language variety (Baker 1992; Garrett 2010). In addition, we aimed to disentangle speaker evaluation and language evaluation, both of which are incorporated in the speaker evaluation paradigm, as many argue that attitudes towards speakers and attitudes towards language are potentially separate constructs (Lehnert 2018; Schoel et al. 2012b). Consequently, we decided to use the Attitudes towards Language Questionnaire (AtoL), which aims to exclusively measure explicit attitudes towards language as opposed to explicit attitudes towards speakers (Schoel et al. 2012b). Our application of the AtoL questionnaire to measure explicit language attitudes was motivated by its careful construction and validation described below. In addition, we aimed to facilitate the contextualisation of our findings, since the AtoL questionnaire has been previously employed in a study investigating language attitudes in Luxembourg (Lehnert 2018).

Overall, the original AtoL questionnaire was developed with carefully conducted statistical analyses, described below, and its validity was confirmed with various cross-linguistic applications, for example, in different speech communities with different samples of speakers (Schoel et al. 2012a). More specifically, the development of the AtoL questionnaire included a principal component analysis of 51 semantic differentials scales taken from previous studies, resulting in the three main factors of language perception represented in the questionnaire: these factors reflect the dimensions of *Sound* (e.g., harsh–soft), *Structure* (e.g., precise–vague) and *Value* (e.g., beautiful–ugly) of a language, towards which participants can hold attitudes. Analyses showed the *Value* dimension to be the superordinate factor of *Sound* and *Structure*. Finally, the construction of the questionnaire included reducing the semantic differentials scales to 15 by analysing the discriminatory power and factor loadings. In the final questionnaire, each of the three factors, *Sound, Structure* and *Value*, has five semantic differential scales with a 5-point scale.

Additionally, the validity and reliability of the AtoL questionnaire was corroborated by its application in measuring language attitudes towards various language varieties in different contexts, e.g., Bavarian, Saxonian, German, English, Chinese (Schoel et al. 2012a). These studies were conducted in different languages of instruction (i.e., German, English, French, Italian, Spanish and Serbian) with diverse samples (including non-student participants). The factors *Value, Sound* and *Structure* were found to account for between 56% and 72% of the total variance in the data of these studies, which corroborates the validity and reliability of the AtoL questionnaire as a new tool for measuring explicit language attitudes.

In further analyses, researchers aimed to contextualise the AtoL questionnaire within previous methodological and theorical approaches to language attitudes (Fiske et al. 2002; Mulac and Lundell 1982; Zahn and Hopper 1985). More specifically, the factor *Sound* was found to be potentially related to the attitude dimension of solidarity (integrative attitudes) (Gardner 1988; Lambert et al. 1968), since measures of warmth (Fiske et al. 2002) and aesthetic quality (Mulac and Lundell 1982) were moderately correlated with this factor. Conversely, *Structure* showed a stronger correlation to competence (Fiske et al. 2002) and socio-intellectual status measures (Mulac and Lundell 1982), indicating that this factor is related to the attitude dimension of status (Gardner 1988; Lambert et al. 1968). Finally, the factor *Value* was intercorrelated with the attitude measures of warmth and language competence, as well as socio-intellectual status and aesthetic quality (Mulac and Lundell 1982). Consequently, Schoel et al. (2012a) argue that the factor *Value* refers mostly to the general overall attitude.

For a good structural fit between theory and methodology, the current study encompasses only the main factor *Value*. We chose the *Value* factor due to the fact that it is superordinate to *Sound* and *Structure*, and it correlates with both attitude dimensions, i.e., status/instrumental attitudes and solidarity/integrative attitudes, as discussed above. Most importantly, the *Value* dimension constitutes a general measure of explicit preference, which is in line with the definition of attitudes adopted here as developed from a social cognitive perspective. In this definition, the fundamental difference between attitudes is based on underlying cognitive processes, namely implicit and explicit, and not the content of the attitude such as the structure or sound of a language variety.

Furthermore, we decided to add one additional semantic differential scale to the five originally included in the *Value* dimension. This additional scale has been previously used in the only study applying the AtoL questionnaire in Luxembourg (Lehnert 2018). More specifically, Lehnert (2018) added one additional semantic differential scale for each dimension, i.e., *Value, Sound, Structure,* in order to adapt the questionnaire for the unique multilingual speech community of Luxembourg.

Overall, our AtoL questionnaire encompassed six semantic differential scales for the *Value* dimension, five from the original questionnaire (Schoel et al. 2012a) and one from Lehnert (2018); see Table 1 below. These six semantic differential scales were combined with three (Belgium) and four (Luxembourg) labels indexing the language varieties under investigation. We selected the labels based on our small-scale (*n* = 19–23)[6] online norming study as well as on previous studies (Entringer et al. 2018; Möller 2017; Neises 2013; Weber 2009). In our norming study, informants were presented with speech samples in the respective varieties (standard and its vernacular) and were asked to provide and chose labels through open and multiple-choice questions on the appropriate name for each variety at issue. In Belgium, the norming study confirmed the two most common designations for the standard and vernacular varieties in the literature, i.e., "Platt" and "Hochdeutsch". In Luxembourg, the norming study showed the same diversity of labels for the standard and its vernaculars as emerged in previous research (Entringer et al. 2018). Example screens in the Appendix A (see Section 5) provide more insights on the labels of the language varieties and the phrasing of the questions which were used in our study.

*2.3. Procedure*

The study was carried out entirely online. Participants were first asked to provide information on their general socio-biographical background and language competence, which took on average 5 min. This was followed by an implicit attitude measure of 15–20 min, reported in Vari and Tamburelli (2020). Finally, explicit attitudes were measured with our AtoL questionnaire, lasting on average 5–10 min. The AtoL questionnaire comprised six semantic differential scales for the *Value* dimension described above. More specifically, participants were asked to indicate on these six scales the positions between six bipolar adjective pairs ranging from 0/left adjective to 4/right adjective, which best described the language variety under investigation. In Luxembourg, the varieties under investigation



were the Moselle Franconian vernacular, German, French and Standard Luxembourgish, while in Belgium, they were the Moselle Franconian vernacular, German and French. The language varieties were presented to participants in this exact order, since the order of presentation remained the same for all trials in each speech community.

All participants evaluated each variety using the same bipolar adjective pairs. Overall, the *Value* ratings on the 5-point scale constituted the dependent variable, while the language variety to be evaluated was the independent variable of the study. The order of presentation of the bipolar adjective pairs was randomised and their positions on the two opposing sides of the semantic differential scales were pseudo-randomised. More specifically, the position of the negative and positive adjectives on either the left or the right side of the scale changed for every 3rd semantic differential scale. The reasons for this was to avoid participants engaging only superficially with the questionnaire, or having their responses influenced by position effects, as both can potentially impact on the validity of responses (Dörnyei and Taguchi 2009).

In Belgium, the language of instruction of the questionnaire was Standard German, while in Luxembourg, participants could choose between either German or Standard Luxembourgish, which reflected the linguistic choice for the Germanic standard languages in each country. The German bipolar adjective pairs were identical to the adjectives used in the original AtoL study (Schoel et al. 2012a) and in the AtoL study previously conducted in Luxembourg (Lehnert 2018). In addition, the German adjectives were translated into Luxembourgish by a native speaker. Table 1 (see below) shows the original bipolar adjective pairs of the AtoL scales plus additions from Lehnert (2018). The *Value* dimension (in bold) is the dimension investigated by the current study.

**Table 1.** Bipolar adjective pairs of the AtoL semantic.

| English | German | Luxembourgish |
|---|---|---|
| VALUE | VALUE | VALUE |
| beautiful–ugly | schön–hässlich | schéin–ellen |
| appealing–abhorrent | ansprechend–abstoßend | uspriechend–ofstoussend |
| pleasant–unpleasant | Angenehm–unangenehm | agreabel–desagreabel |
| inelegant–elegant | unelegant–elegant | net elegant–elegant |
| without style–with style | | |
| clumsy–graceful | schwerfällig–anmutig | schwéierfälleg–liichtfälleg |
| practical–impractical (L) | unpraktisch–praktisch (L) | onpraktesch–praktesch (L) |

Differential scales with additions from Lehnert (2018), here (L).

## 3. Results

Data were screened for duplicates to avoid multiple participation and inverted semantic differential scales were normalised so that 0 always corresponded to the lowest and 5 to the highest possible response. Two participants were excluded from the analysis due to missing responses and contradictory responses for inverted items. The latter are indicative of superficial responding or positions effects, which can potentially impact on the validity of the responses (Dörnyei and Taguchi 2009). The final analysis included a total of 131 participants. All statistical analysis was conducted with SPSS Version 25.

Cronbach's alpha was calculated to ensure internal consistency, i.e., reliability, of the six semantic differential scales for each language variety. The semantic differential scales of our AtoL questionnaire showed a high internal consistency for all language varieties (Cronbach's alpha for all language varieties > 0.734). Accordingly, the proportion of error variance in our AtoL scales was always under 30%.

Before testing our hypotheses, we conducted a multiple ordinal regression analysis to explore the impact of potential confounds such as age, gender and language competence for AtoL ratings as outcome variable. The model coefficients in Table 2 show that neither age nor gender significantly predicted AtoL ratings. Participants' competence in the vernacular showed a trend to significance but overall, Language Variety ($b = 0.23$ and $b = 0.22$, Wald $\chi^2 (2) = 40.97$, $p < 0.001$) and Speech Community ($b = 0.21$, Wald $\chi^2 (1) = 23.92$, $p < 0.001$)

were the strongest significant predictors for AtoL ratings. No interactions of the below predictors reached significance in subsequent analysis and are, therefore, not included in Table 2 below.

**Table 2.** Model coefficients: outcome variable: AtoL_Score; $R^2$ = 0.02 (Cox and Snell), 0.06 (Nagelkerke). Model $\chi^2$ (8) = 75.1, $p < 0.001$ *.

| Predictor | Estimate | 95% Confidence Interval | | SE | Z | *p* | Odds Ratio |
|---|---|---|---|---|---|---|---|
| | | Lower | Upper | | | | |
| age | 0.00423 | 0.01262 | 0.0211 | 0.00860 | 0.492 | 0.622 | 1.004 |
| gender: | | | | | | | |
| male–female | 0.20908 | 0.59500 | 0.1759 | 0.19652 | 1.064 | 0.287 | 0.811 |
| French_competence | 0.21551 | 0.11078 | 0.5433 | 0.16669 | 1.293 | 0.196 | 1.240 |
| Standard_competence | 0.21233 | 0.15370 | 0.5800 | 0.18696 | 1.136 | 0.256 | 1.237 |
| vernacular_competence | 0.30620 | 0.00765 | 0.6213 | 0.16026 | 1.911 | 0.056 | 1.358 |
| speech community: | | | | | | | |
| BELG–LUX | 1.05252 | 0.62855 | 1.4818 | 0.21750 | 4.839 | <0.001 | 2.865 |
| language variety: | | | | | | | |
| Standard–vernacular | 1.43942 | 1.90275 | 0.9838 | 0.23424 | 6.145 | <0.001 | 0.237 |
| French–vernacular | 0.47418 | 0.91824 | 0.0330 | 0.22565 | 2.101 | 0.036 | 0.622 |

We explored the confounding variable of language of instruction in Luxembourg with Mann–Whitney U tests. The AtoL ratings of the vernacular (U = 505; $p$ = 0.439), Standard Luxembourgish (U = 530; $p$ = 0.613), German (U = 485; $p$ = 0.298) and French (U = 462; $p$ = 0.20), were not significantly different in questionnaires in German compared to Luxembourgish.

In order to address research question (a), analyses were guided by hypotheses (1) and (2) (see Section 1.3), which stated that attitudinal differences between Standard German and Luxembourgish would show in (1) relation to their Moselle Franconian varieties and (2) in relation to each other. Consequently, we tested the dependent variable (i.e., AtoL ratings) for (1) within-speech community variation and (2) between-speech community variation. In addition, we investigated research question (b) regarding explicit attitudes towards additional standardised contact varieties in between- and within-speech community analyses.

First, we ran two Friedman's ANOVAs, one on the Luxembourgish and one on the Belgian sample, in order to explore the within-speech community variation. The dependent variable, i.e., AtoL ratings, was not normally distributed in both samples (visual inspection and Shapiro–Wilk $p < 0.001$) and measured on an ordinal scale. Therefore, we preceded with non-parametric tests to investigate the within-speech community variation of AtoL ratings.

### 3.1. Within-Speech Community Analysis: Luxembourg

The non-parametric Friedman's ANOVA in Luxembourg had four levels for the independent variable, i.e., Language Variety (vernacular, French, German and Standard Luxembourgish). Overall, Luxembourgish participants evaluated their language varieties significantly differently ($\chi^2$(3) = 21.97, $p < 0.001$). In addition, we conducted pairwise comparisons—Wilcoxon signed ranked tests with Bonferroni corrections—to further explore the differences in AtoL ratings. Most importantly, the difference in ratings of Standard Luxembourgish vs. its vernacular was significant (z = −4.45, $p < 0.001$), with the vernacular eliciting higher ratings than Standard Luxembourgish. Similarly, AtoL ratings of the vernacular and Standard Luxembourgish were significantly different to German (z = −4.22, $p$ = 0.001, for vernacular and German; z = −3.12, $p$ = 0.002 for Standard Luxembourgish vs. German) and participants' AtoL ratings for the vernacular and German were higher than Standard Luxembourgish.

The differences between the AtoL ratings of French vs. all other language varieties did not prove to be significant, i.e., French vs. vernacular (z = −1.46, *p* = 0.145), French vs. German (z = −0.20, *p* = 0.884) and French vs. Standard Luxembourgish (z = −2.40, *p* = 0.016, non-significant with Bonferroni correction, significance level raised to α = 0.008). Figure 3 summarizes the results for Luxembourg.

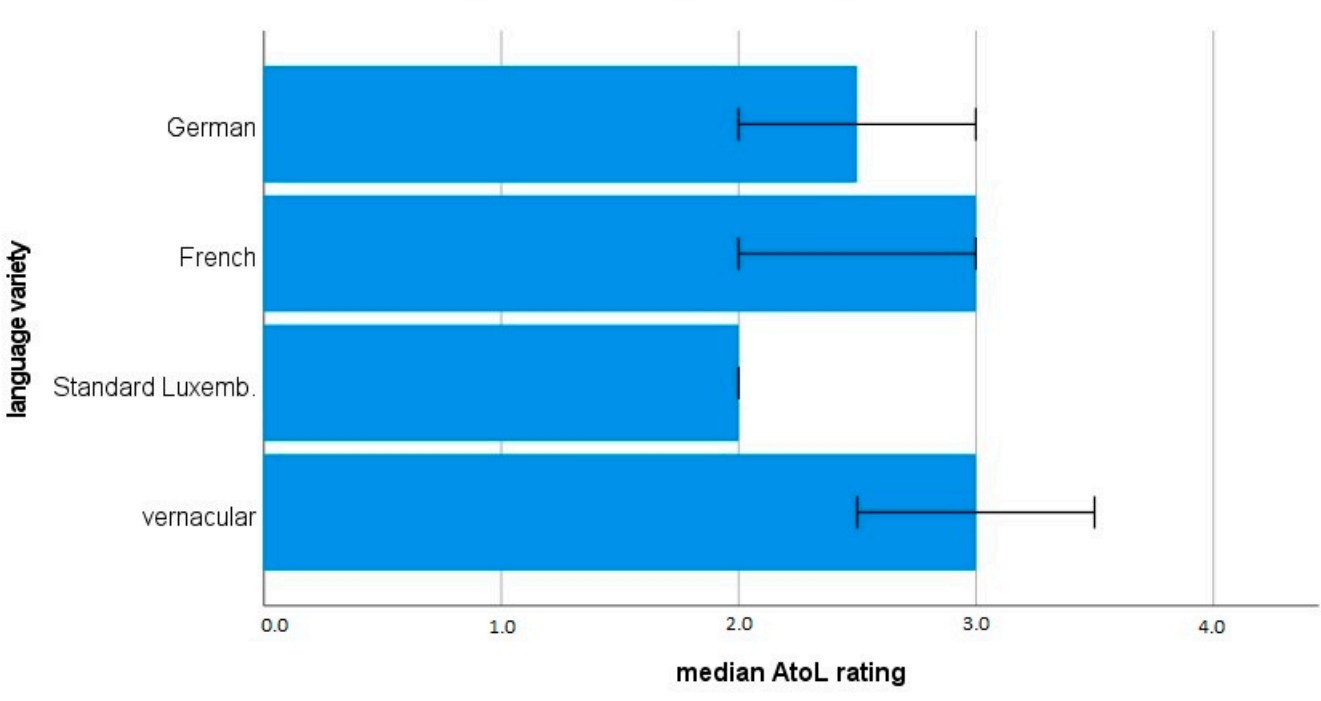

**Figure 3.** AtoL ratings in Luxembourg: vernacular* (median = 3.00; IQR = 2.00, 3.00, 4.00), standard* (median = 2.00; IQR = 2.00, 2.00, 2.50), French (median = 3.00; IQR = 2.00, 3.00, 3.38), German* (median = 2.50; IQR = 2.00, 2.50, 3.00). * sign. different.

### 3.2. Within-Speech Community Analysis: Belgium

We ran a second, non-parametric, Friedman's ANOVA on the Belgian AtoL ratings, but this time with three levels of the independent variable (i.e., Language Variety), namely vernacular, Standard German and French. Overall, Belgian participants rated their language varieties significantly differently on the AtoL scales ($\chi^2(2)$ = 28.79, *p* < 0.001). Pairwise comparisons explored these differences further, revealing that participants evaluated almost all language varieties significantly differently, with both French and the vernacular eliciting higher ratings than Standard German (z = −4.19, *p* < 0.001 for French vs. Standard German; z = −3.92, *p* < 0.001 for the vernacular vs. Standard German). Only the difference in evaluation between the vernacular and French did not prove to be significant (z = −1.06, *p* = 0.291). Figure 4 summarizes the results for Belgium.

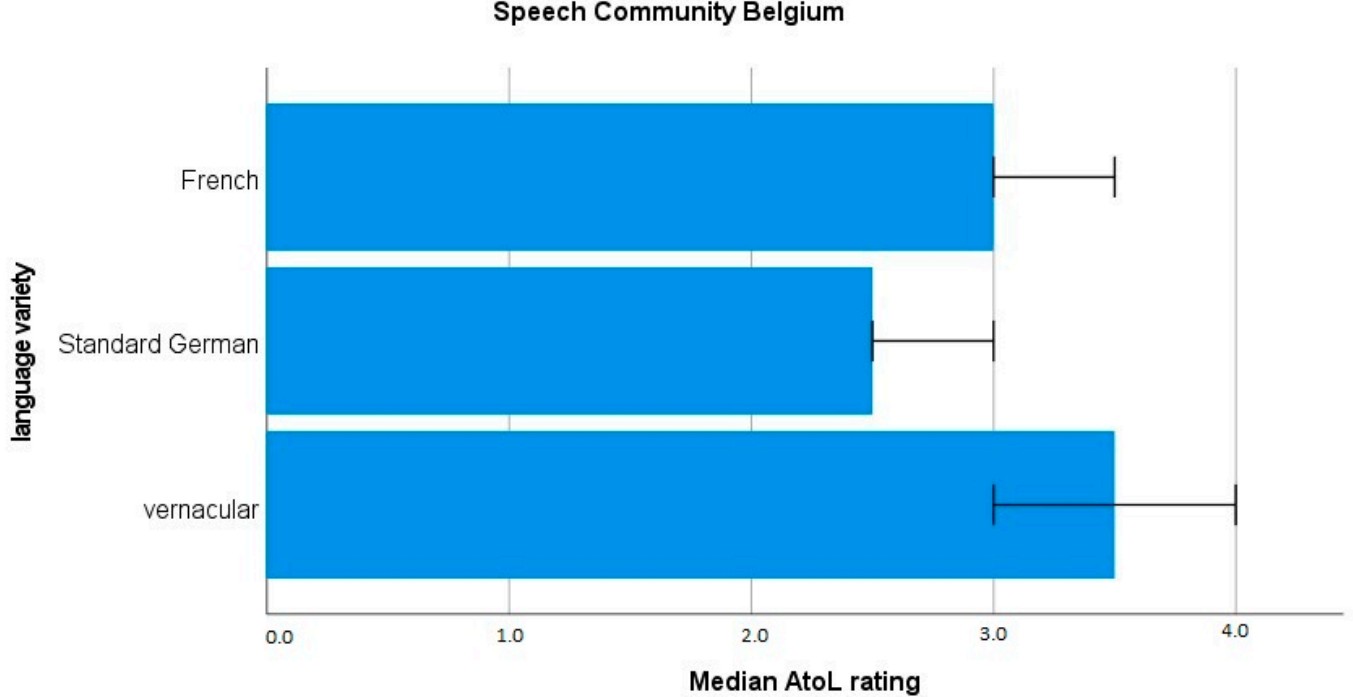

**Figure 4.** AtoL ratings in Belgium for: vernacular** (median = 3.50; IQR = 2.50, 3.50, 4.00), standard* (median = 2.50; IQR = 2.00, 2.50, 3.00), French** (median = 3.00; IQR = 2.50, 3.00, 4.00). * sign. different to all other varieties; ** sign. different to standard variety.

### 3.3. Between-Speech Community Analysis: Belgium vs. Luxembourg

In addition, we analysed the variation of AtoL ratings between speech communities comparing Belgium and Luxembourg. The dependent variable, i.e., the AtoL ratings, was measured on an ordinal scale and not normally distributed. The assumption of equality of variance was not violated except for one cell of the experimental design, i.e., Leven's test for AtoL ratings for French (F(1, 129) = 7.82, $p$ = 0.006). Thus, we proceeded with non-parametric independent samples tests, i.e., Mann–Whitney tests, to analyse the between-community variance of AtoL ratings. The results are summed up in Figure 5.

First, we analysed AtoL ratings for the language varieties present in both speech communities, i.e., French, German and the vernacular, and conducted three Mann–Whitney U tests, with Speech Community as a grouping variable. Luxembourgish participants evaluated French significantly differently from their Belgian counterparts (U = 1545; z = −2.80, $p$ = 0.005). However, the assumption of equality of variance—which is an assumption of the Mann–Whitney U test—was violated for French and in addition, the median AtoL ratings for both speech communities are identical and only the interquartile range is higher for the Belgian AtoL ratings (both medians = 3.00; BELG: IQR = 2.50, 3.00, 4.00; LUX: IQR = 2.00, 3.00, 3.38).

Ratings for German and for the Moselle Franconian vernaculars showed no difference between the two communities (German: U = 1994, z = −0.70, $p$ = 0.485; vernacular: U = 1798, z = 1.63, $p$ = 0.103).

Furthermore, in the second step of the between-speech community analysis, in order to test hypothesis (b) and research question 2, we created two new variables and conducted two further Mann–Whitney U tests. More specifically, we collapsed AtoL ratings for the two standard varieties, i.e., Standard German in Belgium and Standard Luxembourgish in Luxembourg, in a variable called "standard variety". We also created a variable called "crosslinguistic contact variety", which included ratings for French in Belgium (as the

only additional contact variety) and German in Luxembourg as the second additional contact variety. The first Mann–Whitney U test revealed that Luxembourgish participants' AtoL ratings of the standard variety were significantly lower compared to their Belgian counterparts (U = 1361, z = −3.82, *p* < 0.001), suggesting that Luxembourgish speakers have more negative attitudes towards Standard Luxembourgish (median = 2.00), than Belgian speakers have towards Standard German (median = 2.50).

The second Mann–Whitney U test revealed a statistically significant difference between the two groups for the variable crosslinguistic contact variety (U = 1382, *p* < 0.01), suggesting that French in Belgium is associated with more positive attitudes compared to German in Luxembourg.

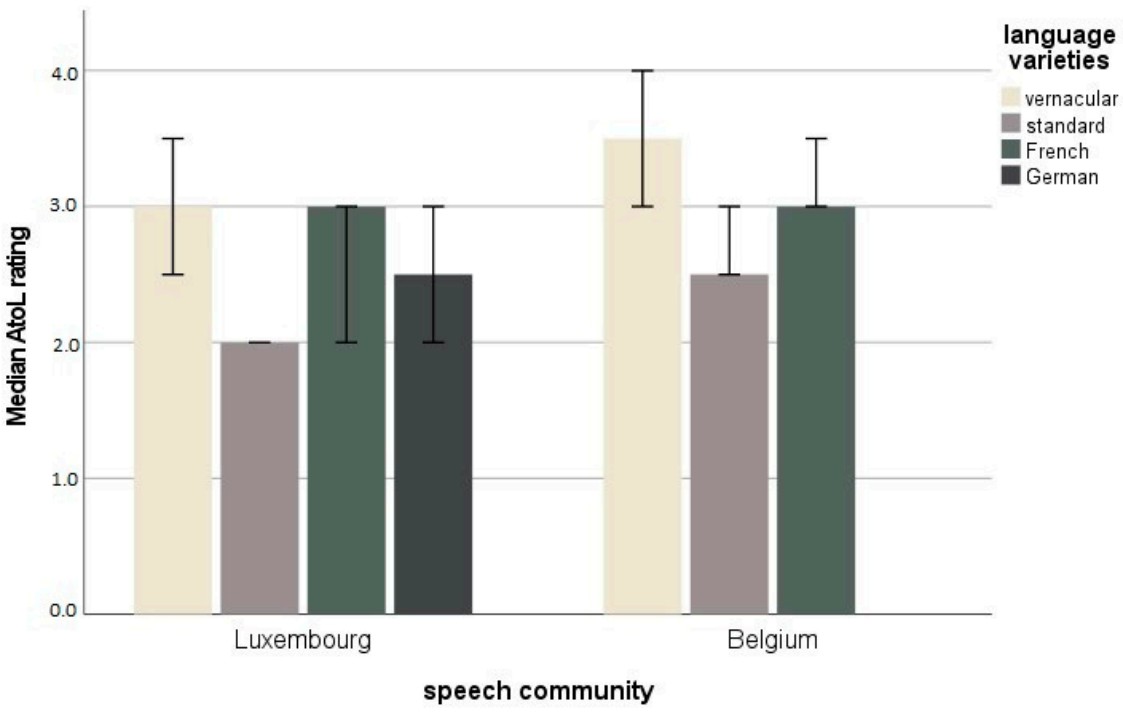

**Figure 5.** Between-speech community comparisons: Luxembourg (LUX) and Belgium (BELG). Vernacular (LUX: median = 3.00; BELG: median = 3.50). standard* (LUX: median = 2.00; BELG: median = 2.50). French* (LUX: median = 3.00; BELG: median = 3.00). German (LUX: median = 2.50; BELG: median = 2.50. Crosslinguistic contact variety* (LUX: German: median = 2.50; BELG: French median = 3.00). * sign. different between speech communities.

## 4. Discussion

Previous studies demonstrated how varieties with varying degrees of standardisation are implemented and accepted differently in (endangered) vernacular speech communities (Ferguson 1968; Haugen 1966; Loureiro-Rodriguez et al. 2013; O'Rourke 2018). In addition, attitudes are found to be indicative of this acceptance (Coupland and Kristiansen 2011; Loureiro-Rodriguez et al. 2013; O'Rourke 2018).

The present study investigated the effect of varying degrees of standardisation on explicit attitudes. Specifically, we explored speech communities in Belgium and Luxembourg, which underwent different standardisation processes resulting in the introduction of different standard varieties for their Moselle Franconian vernaculars, namely Standard German in Belgium and Standard Luxembourgish in Luxembourg. As research has shown the higher degrees of standardisation of German compared to Luxembourgish, we proposed to explore whether explicit attitudes reflect how the respective standard varieties are accepted in their speech communities. We hypothesised that these different degrees of acceptance would surface in explicit attitudes in both within- and between-community comparisons.

In line with our expectations, Luxembourgish speakers evaluated their vernacular significantly more positively than Standard Luxembourgish in within-community analysis. The lower degree of standardisation, which especially shows in a lack of implementation in the educational domains, leads to lower degrees of acceptance of this "new" endogenous standard variety by the vernacular speech community. This result is especially interesting in light of previous attitudinal research in Luxembourg. Previous studies showed mixed results but reported overall positive attitudes towards Luxembourgish, often without distinguishing between Standard Luxembourgish and its Moselle Franconian vernaculars, for example, see Fehlen (2009); Gilles et al. (2010); Lehnert (2018). In addition, they did not focus on the most "non-standard like" vernacular community in Luxembourg, i.e., the Éislek region. Our study investigated this speech community exclusively, and our findings are in line with the only other attitudinal study of this community (Vari and Tamburelli 2020). Both studies found participants to clearly prefer their vernacular over Standard Luxembourgish. This demonstrates the need to distinguish between attitudes towards a standard variety and its vernaculars in the first place and additionally, to include varieties that are more distant from the standard variety in any investigation of the acceptance of a standard variety in vernacular communities.

However, our results did not support the second hypothesis regarding the within-community comparisons. Contrary to what we expected, our results indicate that Belgian participants hold more positive explicit attitudes towards their Moselle Franconian vernacular compared to Standard German. This reflects, to some extent, the findings of the only other study on explicit attitudes in this speech community, which found mainly egalitarian attitudes, but also a preference for vernaculars over Standard German (Weber 2009). In contrast, the findings of a study on implicit attitudes in this speech community showed a preference for the standard variety over the vernaculars (Vari and Tamburelli 2020). Social psychological research provides potential explanations as to why explicit attitudes towards Standard German (in relation to its vernaculars) do not indicate that it is well accepted in the Belgian speech community, despite its high degree of standardisation. Research demonstrates that social desirability often leads to more egalitarian explicit attitudes, or to explicit attitudes that show overcorrected implicit negative biases leading to a preference of the subordinate group in intergroup relationships (Dovidio et al. 2009; Wilson et al. 2000). Accordingly, Belgian participants might have overcorrected their demonstrated implicit negative bias towards their vernacular (Vari and Tamburelli 2020) and consequently reported a clear preference for their Moselle Franconian vernacular in explicit attitudes. The social desirability of attitudes might have touched on attitude contents such as the covert prestige of Moselle Franconian (see Trudgill 1972) and/or the solidarity dimension of attitudes reflecting feelings of belonging (Cargile et al. 1994; Ryan Bouchard et al. 1982). Overcorrection processes of negative implicit biases could have not taken place in explicit attitudes in Luxembourg, since speakers also have a preference for the subordinate variety, i.e., the Moselle Franconian vernacular, when tested on implicit attitudes (Vari and Tamburelli 2020). This would be in line with the post hoc explanation that social desirability in the speech communities involves the covert prestige of Moselle Franconian and its positive evaluation on the solidarity dimension.

For the between-community comparison, we expected Luxembourgish speakers to hold more negative explicit attitudes towards the standard variety compared to Belgian speakers. The results of our AtoL ratings support this hypothesis. Luxembourgish speakers evaluated Standard Luxembourgish less favourably compared to Belgian speakers' evaluation of Standard German. This difference in explicit attitudes towards the respective standard variety is especially noteworthy in light of comparably positive attitudes towards the endangered Moselle Franconian variety in both speech communities. This contrast highlights that attitudinal differences between the speech communities lie in their different explicit evaluation of their respective standard varieties and not in their evaluation of their closely related endangered vernaculars.

Finally, our last research question was exploratory in nature and concerned explicit attitudes towards the other standardised contact varieties in the speech community, namely French in Belgium and German and French in Luxembourg. These additional contact varieties compete with the respective standard varieties over usage in H(igh) domains and could also act as potential functional standards (in the sense of Muljacic 1989; see also Gilles 2019) for the endangered vernaculars. Thus, we suspected that Standard German in particular, which acted formerly as the structural standard variety in Luxembourg, might impede the implementation of the "new" standard variety, i.e., Standard Luxembourgish. Indicative of such an impediment would be very positive attitudes towards Standard German in Luxembourg, showing that the variety still carries prestige in the speech community. In contrast, previous research in Luxembourg seemed to show negative attitudes towards German, despite its still widespread usage (Fehlen 2009, 2016; Gilles et al. 2010). However, these attitudinal studies did not distinguish between Standard Luxembourgish and vernacular Moselle Franconian varieties, collapsing them together under a generic "Luxembourgish". Our study filled this research gap by investigating attitudes towards the vernacular and its "new" standard, i.e., Standard Luxembourgish, and how these fare in relation to attitudes towards Standard German. However, lacking previous attitudinal research, we were unable to present a hypothesis for this investigation. Notably, we found that vernacular Moselle Franconian speakers hold significantly more positive attitudes towards German compared to Standard Luxembourgish. This might be indicative of Standard German impeding the acceptance of Standard Luxembourgish in this speech community. Once again, our findings highlight the need to take into account the internal variation of endangered languages, specifically the differences between the endangered vernacular and its associated standard, warning against making a priori assumptions of homogeneity.

Our last research question also referred to French, which is present as an additional contact variety in both speech communities. However, the evaluation of French did not differ significantly from any other language varieties in between-speech community analyses. In within-speech community analyses, French differed only from Standard German in Belgium, where it was evaluated more positively. These findings are in contrast with previous suggestions of a limited influence of French as an additional standard variety in the *Belgische Eifel* region of Belgium (Darquennes 2019). In addition, cross-linguistic analysis showed that French was evaluated more positively in Belgium compared to German in Luxembourg. The lack of significant differences in other comparisons could again be indicative of the influence of social desirability on egalitarian attitudes. However, previous research in Luxembourg reported very favourable explicit attitudes towards French, occasionally even more favourable than towards Luxembourgish (Fehlen 2009; Gilles et al. 2010; Lehnert 2018). In contrast, the only study including implicit attitudes found a preference of Luxembourgish over French (Lehnert 2018). The contradictory findings in Luxembourg and the lack of attitudinal research in Belgium make it difficult to contextualise our findings. Overall, more research is needed on attitudes towards the standard variety and its vernaculars in relation to additional varieties in order to fully understand standardisation in language contact situations.

Two caveats of our study are a potential selection bias and order effects. First, our participant recruitment via media and local societies such as women's clubs might have led to the recruitment of participants with a particular interest in the vernacular speech community. In addition, our findings could be influenced by order effects, since—due to technical issues—the order of presentation remained the same for all trials in each speech community. However, both practices are common in language maintenance research, where participant recruitment is potentially biased towards including predominantly "language enthusiasts" from the local community, particularly—albeit not solely—in cases where the overall number of speakers is low (e.g., Deminger 2000). Elderly speakers of endangered languages, who require paper versions of questionnaires, are also commonly subject to order effects in these studies.

To summarise, our comparative study found indications of an incomplete standardisation of Luxembourgish that results in more negative explicit attitudes. This standardisation might be potentially impeded by the former standard variety, Standard German, which still carries prestige in the community. Attitudes towards the standard variety of Belgium, i.e., Standard German, reflected its higher degrees of standardisation and acceptance only in comparison to Standard Luxembourgish, not in comparison to the Moselle Franconian vernacular. We suggested that this explicit preference of the vernaculars in Belgium might be due to overcorrection processes of implicit negative biases against the vernacular (Vari and Tamburelli 2020). These overcorrection processes could be based on the socially desirable attitude dimension of solidarity and/or reflect the covert prestige of Moselle Franconian in Belgium. Unfortunately, our study could not investigate social desirability and the attitude dimensions of solidarity vs. status, since it lacked insights from previous studies in the speech communities in order to advance any hypotheses. Future research needs to explore this avenue further.

## 5. Conclusions

Overall, this study showed the importance of considering the internal variation within an endangered language by distinguishing between endangered vernaculars and their standard varieties when measuring attitudes. Similarly, the study also showed that research into the acceptance of a standard must include speakers of vernaculars that are distant from the standard at issue. In addition, our study showed that a complete understanding of the potential obstacles that may impede the acceptance of a standard variety in an endangered speech community must include exploration of the attitudes towards the standard and its vernaculars in relation to other standardised contact varieties.

Most importantly, our study indicates that there are potential limitations to relying on standardisation as a language maintenance effort. As a cautionary note to the widespread belief that the introduction of a standard variety will necessarily bolster attitudes towards an endangered vernacular (Fishman 1991; Lewis and Simons 2010) and that a standard variety will complement endangered vernaculars with its prestige (Fishman 1991), our study has shown that only a fully accepted standard variety carries the prestige that can potentially positively influence endangered vernaculars. Our results in Luxembourg suggest that a newly introduced standard variety may sometimes not yet fully carry the prestige that would be needed to have a strong positive effect in the endangered speech community. Therefore, the introduction of a standard variety might not contribute to reversing the loss of the endangered vernaculars until the standard is fully accepted. However, other factors might also contribute to the success of standardisation as a tool to bolster endangered vernaculars. For example, researchers have identified the role of linguistic distance between the standard and its endangered vernaculars as well as the types of attitudes (implicit vs. explicit) as potentially limiting the positive effect of a standard variety's prestige (Vari and Tamburelli 2020).

**Author Contributions:** Conceptualization, J.V. and M.T.; Formal analysis, J.V.; Investigation, J.V.; Methodology, J.V.; Project administration, J.V.; Supervision, M.T.; Visualization, J.V.; Writing—original draft, J.V.; Writing—review & editing, J.V. and M.T. Both authors have read and agreed to the published version of the manuscript.

**Funding:** This research received no external funding.

**Institutional Review Board Statement:** The study was approved by Ethics Committee of Bangor University (protocol code SLLL-026, 11 June 2019).

**Informed Consent Statement:** Informed consent was obtained from all subjects involved in the study.

**Data Availability Statement:** The scripts for the questionnaires and further stimuli can be found on the Open Science Framework in the project folder DOI 10.17605/OSF.IO/E8NMP.

**Conflicts of Interest:** The authors declare no conflict of interest.

## Appendix A

Example screens of AtoL questionnaires with German as the language of instruction

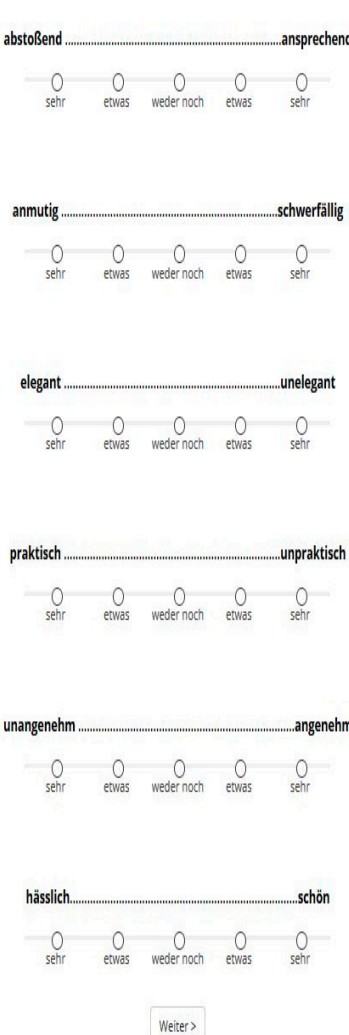

**Figure A1.** Semantic differential scales to measure attitudes towards the vernacular in Belgium.

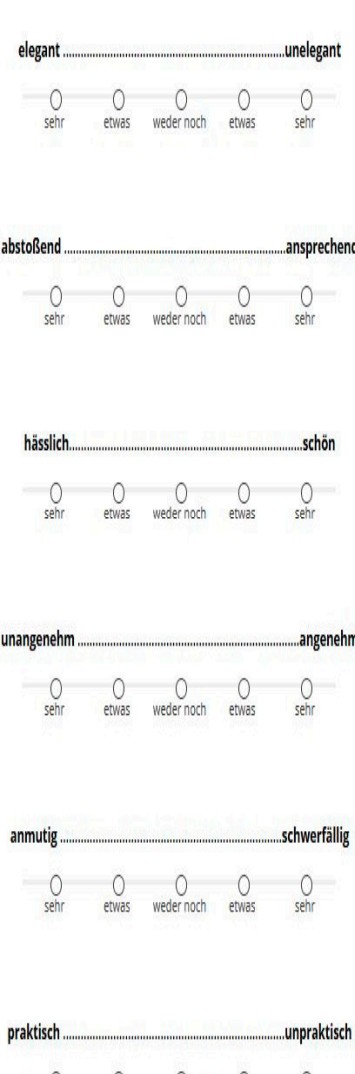

**Figure A2.** Semantic differential scales to measure attitudes towards the vernacular in Luxembourg.

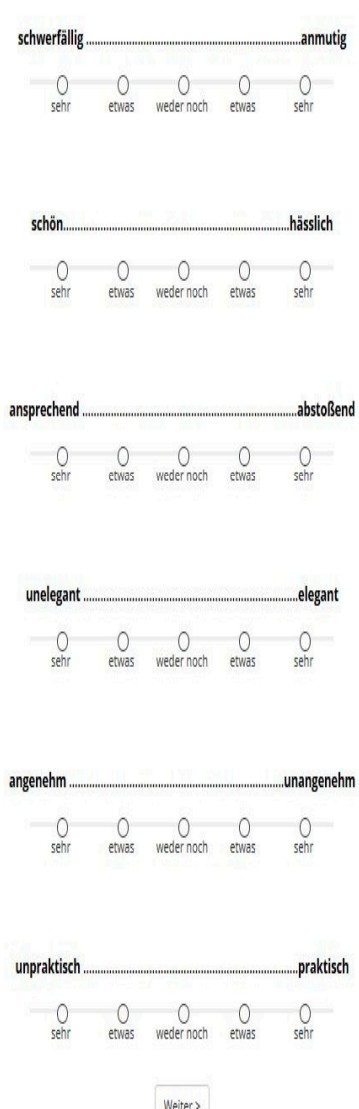

**Figure A3.** Semantic differential scales to measure attitudes towards Standard Luxembourgish in Luxembourg.

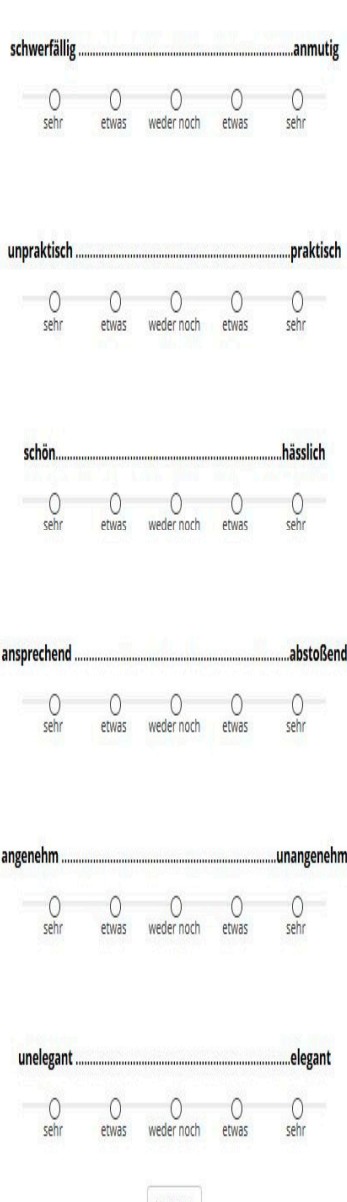

**Figure A4.** Semantic differential scales to measure attitudes towards German in Belgium and Luxembourg.

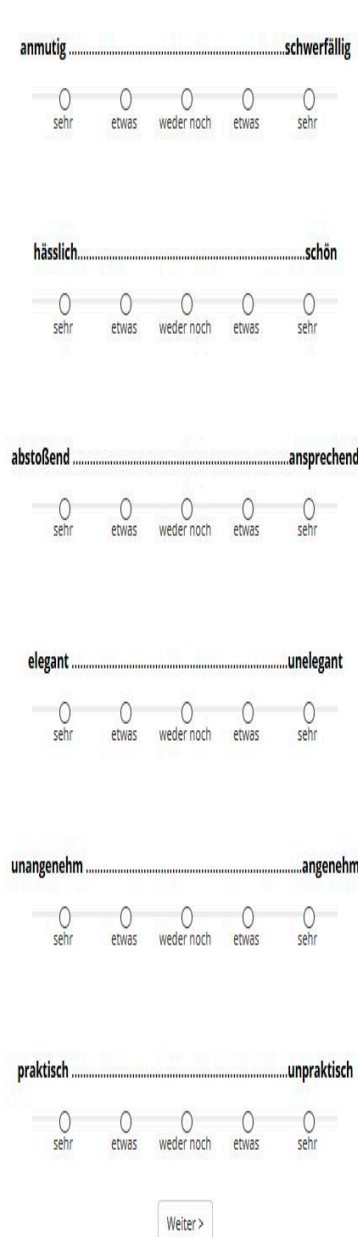

**Figure A5.** Semantic differential scales to measure attitudes towards French in Belgium and Luxembourg.

## Notes

[1] While we will use this denomination in our presentation, the object of our study is Moselle Franconian, a Germanic vernacular spoken in the *Deutschsprachige Gemeinschaft* and not "German".

[2] Most famously, Muljacic (1989) defines a functional standard variety in opposition to a structural standard variety. A functional standard is genetically unrelated or only very distantly related to the vernaculars in the speech community but has standard functions in relation to them.

[3] Limburgian-Ripuarian is also identified as vulnerable by UNESCO, but it is Moselle Franconian we are concerned with here.

[4] The language law in 1984 did not use the term "official language", but it defined Luxembourgish to be the national language, next to German and French, as the languages of administration and judiciary (see Fehlen 2016).

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
