# Peer review of "Accepting a “New” Standard Variety: Comparing Explicit Attitudes in Luxembourg and Belgium"

_languages, doi:10.3390/languages6030134_

Round 1

Reviewer 1 Report

Review report Accepting a “new” standard variety: Comparing explicit attitudes in Luxembourg and Belgium

This paper reports on a study focussing on explicit language attitudes in two areas with endangered vernaculars, namely the Moselle Franconian vernaculars in Northern Luxemburg and Eastern Belgium. Although the topic of the paper is most certainly worthy of study, I do find that the study has some important issues that need to be addressed before I can recommend it for publication. Below I first discuss these main issues, after which I list some additional issues and questions per section of the paper.

Main points:

  • Literature review lacks structure and needs to be more succinct. Especially the sections 1.1 and 1.2 lack internal structure and could do with being much more concise. Section 1.3 could also be more to the point. The discussion likewise contains a lot of repetition and could be condensed significantly.
  • “These make an ideal testing ground for an investigation of how different degrees of standardisation resulting in different degrees of acceptance may surface in different attitudes across two speech communities.” As well as section 1.3: I disagree with this narrow interpretation of the relationship. There may be many reasons why language attitudes differ between the communities. This is not a controlled experiment where you can point to one factor that is responsible for the difference between two samples. Although what the authors suggest is a very interesting idea, they should be much more nuanced about it.
  • It seems to me that this study was not designed as a study in its own right, but that it is part and parcel of a larger study that combines an implicit and explicit attitude measurement (see lines 555-556). I feel the study should have been reported as a whole which may have led to a more insightful paper.
  • I don’t feel the authors’ choice to drop two major dimensions of the AtoL instrument is justified. This links up with my major concern that this study was not designed as a study in its own right. It may have been a justifiable choice in the context of the study as a whole, but if the authors want to report on part of the study in a paper in its own right, the explicit attitude measurement should stand up on its own which I feel it does not.
  • Lines 583-585: “Due to technical issues, the order in which the language varieties were presented to participants to be evaluated, remained the same for all trials in each speech community.” à This seems like a major issue to me. Please report what order the varieties were shown in and take this into account in your discussion of the results, as it may severely impact the outcome of the study

Additional points:

LITERATURE REVIEW:

  • The discussion of standardisation in relation to language attitudes misses some crucial work, like Grondelaers et al. (2016).
  • As the study revolves around the idea that introduction of a standard language can support an endangered language, this mechanism could be discussed in much more detail.
  • The relation between standard varieties and prestige is presented in a circular way in the paper. Sometimes the authors seem to suggest that a variety is standardised by virtue of its prestige, but then at other times it’s suggested that its standard status is also invoked as an explanation for its prestige.
  • Lines 148-151: the reader is left wondering which variety was ranked first
  • Lines 159-170: it should be noted that the opposite pattern has also been well documented in the literature. Take the work by Kristiansen that shows explicit preference for Standard Danish compared to the Copenhagen variety, but indirect attitude measures point to the reverse.
  • Figure 1 and 2: some parts of the images seem distorted
  • Description of the Belgian and Luxembourg regions and there varieties could be more succinct and more clearly structured
  • Lines 420-422: “Social psychological research suggests that only the investigation of both attitude types, explicit and implicit, reflects how well a standard variety is accepted in an endangered speech community”. à Please provide references for this statement
  • Lines 426-428: “Some studies find explicit attitudes to be the “window into the future”, arguing that attitude change manifests first in explicit attitudes (Dovidio et al., 2009; Wilson et al., 2000).” à and obvious reference missing here is McKenzie & Carrie (2018)

METHODOLOGY:

  • Please give a clearer explanation why 36 participants were removed from the sample
  • Table 1: it seems superfluous to includes scales that were not part of the study
  • Why were participants asked for information on language competence before they started the attitudinal measures? This does not seem like the best choice, as it may have influenced their behaviour in the subsequent tasks.
  • Is language choice of the Luxembourg participants taken into account in the analysis?
  • Please discuss the labels chosen to refer to the different language/varieties in the study. How were these decided on? Were pretests involved?

ANALYSES:

  • Please provide error bars on both bar plots
  • Figure 5: why are there four bars for Belgium, given that only three language varieties were included in study for that region?

GENERAL:

  • A strong point of this study is that it focuses on communities that are less well studied in sociolinguistic attitude research.
  • The reference list contains Lehnert et al. 2018a and Lehnert et al. 2018b, but several of the Lehnert et al. (2018) references throughout the article are neither marked as a or b, which is rather confusing
  • The numbering of tables and figures seems to have gone wrong
  • Line 593: in à remove .
  • Line 860: standard, r warning à remove r

Reviewer 2 Report

Comments to

Accepting a “new” standard variety: Comparing explicit attitudes in Luxembourg and Belgium

GENERAL

I liked to read this paper, and it has a number of merits (especially its pioneering work in an understudied area across national borders), but there are a number of grave concerns which have to be addressed in my view before this can be published.

To begin with, quite a lot of essential information is missing to get a reliable view of the data:

  • What is your definition of “implicit” and “explicit” attitudes? This does not really become clear, and it is absolutely essential to gauge the theoretical leg-up to your study, but also the design of your own study.
  • What linguistic stimuli did you present? Were they labels, or audio samples? Please provide much more info.
  • For all the other issues on this point, see the detailed comments below

Second, I have some grave concerns about the set up and the execution of the study:

  • Please describe, as explicitly as possible the order in which the stimulus varieties were presented. I admire your honesty in admitting that you could not rotate them, but please describe in as much detail as possible what the impact of this order is. Could the low prestige you find for Standard German in Luxemburg, for instance, be related to its rank in the order of presentation? More crucially, can you exclude this?
  • Why didn’t you include Respondent Age and Gender in your statistical analysis? In view of the fact that you’re investigating an endangered language, I would say that it is pivotal to see how your younger respondents react to it, no?

Finally, I am not convinced that the AtoL questionnaire is the most appropriate tool to investigate what you want to find out

  • You indicate that research has correlated Value with prestige perceptions, but in order to study standard language dynamics, I believe that it is much more useful to investigate different sources of prestige (conservative and modern), which have been found to impact dynamics
  • I cannot really infer from what you write what your problem with a traditional speaker evaluation experiment with audio samples and speaker personality scales is: in my view, this would be much more suited to your purposes – but I’m sure that there are reasons which I have not understood well.

DETAILS.

Abstract                              “Language maintenance efforts aim to bolster attitudes towards endangered languages by providing them with a standard variety”

Please unpack. I can see why language maintenance could profit from providing a language with a standard, but what is the role of the attitudes in this process? Are people encouraged to believe that there is a standard? Or is the standard provided first?

Abstract l 8                         Why past tense “suggested”?

Abstract ll11-20                 I’m sorry but I can’t follow. How can a vernacular (Moselle Franconian) benefit from the standardisation of the national languages in the communities in which it is spoken (Luxembourgish and German)?

P2/59-64                             I am at a loss what you mean here: unless you specify more clearly how an endangered vernacular could benefit from the standardisation of the norm languages in the communities in which it is spoken. What am I missing?

P2/66                                    Unless you clarify the purpose, the term “research gap” sounds like a straw man. I believe it would be easier if you presented the case of Moselle Franconian first, and then introduce the theoretical notions which now confuse me.

P3/81-82                             “whereas in Belgium, the Moselle Franconian speakers associate their vernaculars with Standard German”

                                               Isn’t Standard German the standard which endogenously developed from Moselle Franconian? Was it imported as an exogenous variety? How can one “associate” one’s vernacular to a standard if there is no genetic relation?

P4/122-124                         “Consequently, attitudes towards Standard German are shown to be overwhelmingly positive when compared to other standardised majority languages (Rothe, 2012; Schoel, Eck, 124 Roessel, & Stahlberg, 2012)”

                                               Please unpack. Are you referring to other majority languages in Germany?

P5/138                                 Speakers’ perception of what?

P5/144                                 This is the fourth time or so I come across the specification “explicit” preceding “attitudes”. This specification begs the question what exactly you mean by the term, and how you contrast it with implicit attitudes. Are you preluding to the fact that your instrument – a questionnaire – is incapable of extracting implicit attitudes? Please specify. Do you need the specification “explicit” at all at this stage?

                                               OK, while reading on I have the impression that you are referring to attitudes collected in speaker evaluation-like settings, no? There is some debate as to how explicit these are. To proponents of Implicit Association tasks, like Laura  Rosseel, they would be explicit. To Tore Kristiansen and myself, they would be implicit, if at least you succeed in keeping your participants ignorant of your research ambition.

P5/155-156                         “Additionally, the study found somewhat positive attitudes towards Standard German on the status/instrumental attitude dimension”

                                               If attitudes are only “somewhat positive towards Standard German on the status … dimension”, I don’t understand your claim I refer to above (P4/122-124). Belgian Dutch, to give an example, is always and invariable deemed the more superior variety, at whatever level of consciousness you measure attitudes towards it.

P5/160-161                         Please specify how these authors measured implicit attitudes.

P5/166-167                         This goes against the grain in view of the fact that implicit attitudes are typically more appreciative of standard varieties. Laura Rosseel’s work on Belgian Dutch (2019 in the Implicitness theme issue) furnishes a counter example which is interesting for your work: she found that Belgian Colloquial Dutch was extolled more on the explicit than on the implicit measures.

P6/175-176                         How can a standard (French) be a functional standard variety of originally German vernaculars?

P9/269                                 But this was the implicit attitudes study, no? Can you compare that to the other studies?

P12/336-343                      I find it increasingly difficult to disentangle your use of explicit and implicit, speaker evaluation and language evaluation. Language evaluation typically bypasses participant awareness by relying on speaker evaluation – as you know. But it is possible to include both in one speaker evaluation experiment which has clearly linguistic purposes. A case in point is

Grondelaers, Stefan, Roeland van Hout and Mieke Steegs (2010). Evaluating regional accent variation in Standard Dutch. Journal of Language and Social Psychology 29, 101-116.

Additionally: I know Kristiansen (2011) quite well, and I cannot recall that he had anything to say about the explicit/implicit discrepancy.

P12/345                               Could you please omit some instances of “to the best of our knowledge”? Your literature review is impressive, doesn’t need this hedging ?.

P12/348                               What do you mean by “conflating”?

P13/368                               I have little experience with language maintenance work, but can speech communities themselves be endangered? I’d think it’s their vernacular which is under pressure, no?

P13/373                               Deumert & Vandenbussche (2003) is an overview of European standardisations: it does not foster any evidence about the point you are making here.

P13/387-389                      I don’t see how an investigation of attitudes towards Standard German and Standard Luxembourgish is going to help you settle your (more basic) question about the status of Moselle Franconian.

P14/420-422                      Again, I don’t completely follow this argument. While (more) implicit attitudes are more indicative of attitude change towards non-standard varieties, it is possible with a completely explicit tool to harvest changing attitudes/ideologies. A case in point is:

                                               Grondelaers S, Speelman D, Lybaert C, and van Gent P. (2020) Getting a (big) data-based grip on ideological change. Evidence from Belgian Dutch. Journal of Linguistic Geography https://doi.org/10.1017/jlg.2020.2

P15/426-427                      I can cite as many other references which suggest that it is implicit attitudes which are the window into the future, as these reflect attitudinal change rather than stable, commonsense explicit attitudes. Please check the studies in Rosseel and Grondelaers (theme issue of Linguistics Vanguard), please check studies in the SLICE research endeavour, which all, or nearly all suggest that it is the more implicit attitudes which have explanatory value for language change, not the explicit. Again, a lot depends on how you define implicit and explicit.

P15/438-439                      The study by Zoe Adams in Rosseel and Grondelaers (2019, Implicitness theme issue) directly addresses this matter. Please also check out the work of Robert McKenzie. The seminal work by Tore Kristiansen essentially relies on a comparison of explicit and (more) implicit attitudes.

P16/478-480                      Please specify on the basis of which criterium you excluded which participants. Be as explicit as possible. Also: your participant recruitment worries me somewhat as it is bound to recruit people who have strong opinions about linguistic matters. Is there a way to check this?

P17/521-522                      What do you mean by “superordinate”?

P19/577-580                      How were language varieties presented, as labels, or as audio clips?

P19/583                               Was that the only independent variable? Certainly you also included Gender and Age, no? Hm, it appears you haven’t.

P19/583-585                      Please specify the order in which the varieties were presented. The absence of randomisation is rather worrisome in this respect.

P21/617-621                      Have you considered factor analysis? This is quite customary in sociolinguistic & social psychological attitude studies. One big advantage of factor analysis or principal component analysis is that it will confirm whether your underlying factor structure matches the factors in function of which you included scales (this need not automatically be the case).

PP22&23                             Why haven’t you included demographic features (Gender & Age) in your analysis? I’d say that especially age would be highly relevant when you investigate changes in the standard status of your vernacular, no?

P25/724-726                      I cannot infer the higher Belgian preference for French from your Figure 5.

PP21-25                               I can see why you would use the AtoL questionnaire in view of its validation across communities, but by ignoring prestige evaluations, you miss out on a pivotal standard language change determinant, viz. Dynamism/modern prestige. If non-standard varieties standardize in modern European communities, it is almost always on account of this modern prestige. Auer (2017) refers to “neo-standards” in this respect, I refer to “emergent standards”. It would have been highly interesting to find out whether any of the investigated varieties have this modern prestige.

P28/814-817                      It in view of especially this point – covert prestige & solidarity evaluations – that I would have opted for a speaker evaluation experiment.

P29/854-856                      Where was Standard German in the order of presentation of your varieties – which you admitted you could not vary in view of technical reasons?  

Reviewer 3 Report

This is a well-written and well-researched study with an interesting design and some clear conclusions. As such it is publishable with some minor revisions.

My main concerns are two-fold. First of all, the role and German and French on the labour market may need some more explicit attention. Are many people from the Belgian Eifel region employed in nearby Germany, where the Aachen region is a powerful attractor? How about Luxemburg? What language used used in the work places of the region studied?

Second, was the sample of speakers recruited perhaps biased towards people with strong regionalist ideologies and hence higher evaluation of the vernacular? This issue also needs to be addressed?

Reviewer 4 Report

GENERAL COMMENTS:

In spite of all of the positive points listed above, I have some suggestions for improvement:

-First of all, this is an extensive paper that could be made more compact.  It contains quite a lot of repetition.  So please screen the paper on redundancy. 

-The design of the case study and the statistical data processing is carefully accounted for.  However, I still have a question with respect to the selection of the so-called VALUE-dimension. The questionnaire for the present paper on explicit attitudes excluded the ‘sound’-dimension and the ‘structure’-dimension from the AtoL-questionnaire and therefore focuses exclusively on the ‘value’-dimension.  Apart from the statement that this factor is  “the superordinate factor of Sound and Structure” (line 512), we do not learn what this dimension really stands for and in what respect it actually differs from e.g. the ‘sound’ dimension.  It is hard to understand for the uninformed reader why e.g. “pleasant-unpleasant” is a pair that clusters in the value dimension and not in the sound dimension. 

-Also with respect to the design of the case study: I miss information on the social profile of the respondents (average age, level of education).

-Beware of circular argumentation: you describe the Luxembourgian linguistic situation and point out that “Standard Luxembourgish has not reached the last stage of full implementation” (line 232), whereas other standard languages that are at issue here are fully implemented in H-domains…  And then you hypothesize that fully fledged standard languages will elicit more positive attitudes than standard languages that are not fully fledged.  The elicited attitudes obviously inform us on the acceptance of the standard languages at stake, but I’d be careful not to suggest that lower appreciation points to lower implementation, because then the circle is complete (see e.g. line 794: you wonder why explicit attitudes towards Standard German do not indicate that it is well implemented).  Moreover, I do not know whether the bipolar adjective pairs presented to the respondents warrant conclusions on the implementation of the language variants.

-With respect to the final conclusion: The discussion section is quite long.  I’d separate ‘discussion’ from ‘conclusion’.  In the conclusion I’d include the final paragraph of the present discussion (lines 893 to 901).  That seems like the real conclusion to me.  However, what I miss there is a reference to the finding that the vernacular variant appears to carry more prestige than the fully fledged standards.  That makes one wonder: do you really need the development of (fully fledged) standards for reversing language shift/loss?  I’d like some reflection on this.  Apart from that I think you can compress the discussion section.  E.g. do you have to repeat the hypotheses in full?  You could simply state: contrary to what we expected… / or: in line with our hypotheses

-Example screens in the Appendix: Is it necessary to include so many repetitive example screens in the appendix, just to make clear what labels were used for the language varieties?

MINOR REMARKS:

-lines 58-61: “The prestige of vernaculars….traditional overt prestige tied to status and domination”: I do not see how ‘status and domination” are linked to vernaculars  (maybe the way things are presented here is somewhat confusing)

-line 69: there seems to be something wrong with the syntax here, or maybe “thus prestige” has to be put between commas or dashes?

-line 91: “vulerable”: typo

-line 130: processes: why plural?

-line 149: “research suggests”: should be between commas or dashes

-line 159-160: I’d use a past tense for ‘prefer’ and ‘like’

-line 198: “varieties” = plural as opposed to “Moselle Franconian” >> add “vernaculars”

-line 213: title of section 1.2: not elegant

-line 250: “of the context” >> line 253: “at a contextual level”: repetition

-line 265: “than TOWARDS varieties” (I’d add ‘towards’)

-lines 285-288: strange/confusing sentence: The Moselle Franconian of the Eislek region is closely related to the vernacular of the most northerly part of the Eislek ??? (I’d say: of course it is)

-lines 335-336: repetition of lines 177-179

-lines 521-522: which data are you referring to? Source?

-line 553: see table 3: should be table 1

-line 555: reference to a small-scale norming study: was not introduced before.  What was the goal to that norming study.  What does (n = 19-23) stand for?  19 respondents for…?  23 respondents for…?

-line 581-582: “the order…evaluated”: should be deleted?  Seems like a remnant of a previous version?

-line 617: “in in”: typo

-line 669: remove commas

-lines 711-712: font should be bigger

-lines 832-833: repetition of ‘despite’

-line 849: why a full stop before ‘where’?

-lines 870-871: I do not see the point here: one can provide several paper versions with different orders?

-line 874 (and elsewhere): Can Standard German really be presented as the ‘former’ standard variety?  It still functions as standard variety in e.g. school contexts.

-line 888: something wrong with the wording

Round 2

Reviewer 2 Report

I congratulate the authors on their thorough revision of their paper, and on its current quality. There are a number of points which I continue to find problematic - such as the absence of rotation, which they can't remedy - and the assessment tool they use, but they have been able to clarify their theoretical purpose and methodology to such an extent that their paper represents a truly valuable contribution to the field. I wish them luck in their further work.

Author Response

Thank you very much again for the very insightful comments and literature recommendations!